# SARS-CoV-2-specific CD4+ and CD8+ T cell responses can originate from cross-reactive CMV-specific T cells

Cilia R Pothast[1]*, Romy C Dijkland[1], Melissa Thaler[2], Renate S Hagedoorn[1], Michel GD Kester[1], Anne K Wouters[1], Pieter S Hiemstra[3], Martijn J van Hemert[2], Stephanie Gras[4,5], JH Frederik Falkenburg[1], Mirjam HM Heemskerk[1]

[1]Department of Hematology, Leiden University Medical Center, Leiden, Netherlands; [2]Department of Medical Microbiology, Leiden University Medical Center, Leiden, Netherlands; [3]Department of Pulmonology, Leiden University Medical Center, Leiden, Netherlands; [4]Department of Biochemistry and Chemistry, La Trobe Institute for Molecular Science, La Trobe University, Victoria, Australia; [5]Department of Biochemistry and Molecular Biology, Monash University, Clayton, Australia

**Abstract** Detection of severe acute respiratory syndrome coronavirus 2 (SARS-CoV-2) specific CD4+ and CD8+ T cells in SARS-CoV-2-unexposed donors has been explained by the presence of T cells primed by other coronaviruses. However, based on the relatively high frequency and prevalence of cross-reactive T cells, we hypothesized cytomegalovirus (CMV) may induce these cross-reactive T cells. Stimulation of pre-pandemic cryo-preserved peripheral blood mononuclear cells (PBMCs) with SARS-CoV-2 peptides revealed that frequencies of SARS-CoV-2-specific T cells were higher in CMV-seropositive donors. Characterization of these T cells demonstrated that membrane-specific CD4+ and spike-specific CD8+ T cells originate from cross-reactive CMV-specific T cells. Spike-specific CD8+ T cells recognize SARS-CoV-2 spike peptide FVSNGTHWF (FVS) and dissimilar CMV pp65 peptide IPSINVHHY (IPS) presented by HLA-B*35:01. These dual IPS/FVS-reactive CD8+ T cells were found in multiple donors as well as severe COVID-19 patients and shared a common T cell receptor (TCR), illustrating that IPS/FVS-cross-reactivity is caused by a public TCR. In conclusion, CMV-specific T cells cross-react with SARS-CoV-2, despite low sequence homology between the two viruses, and may contribute to the pre-existing immunity against SARS-CoV-2.

*For correspondence:
c.r.pothast@lumc.nl

**Competing interest:** The authors declare that no competing interests exist.

## Editor's evaluation

This study examines the possibility that high incidence of SARS-CoV2 reactive T cells occur in apparently COVID 19 naive individuals. The study sheds new light on how unrelated virus-specific T cells might be involved in generating immunity to SARS-CoV-2 and is an elegantly performed study.

## Introduction

The effectiveness of the innate and adaptive immune system is an important factor for disease outcome during infection with severe acute respiratory syndrome coronavirus 2 (SARS-CoV-2; *Brodin, 2021*). CD4+ and CD8+ T cells are important components of the adaptive immune system as CD4+ T cells promote antibody production by B cells and help cytotoxic CD8+ T cells to mediate cytotoxic lysis of SARS-CoV-2 infected cells (*Sette and Crotty, 2021*). Whilst immunity is commonly measured solely based on antibody titers, research into coronavirus disease 2019 (COVID-19) pathophysiology and vaccination effectiveness has associated an effective T cell response with less severe COVID-19

(*Bange et al., 2021*; *Bertoletti et al., 2021*; *Liao et al., 2020*; *Rydyznski Moderbacher et al., 2020*; *Sekine et al., 2020*; *Sette and Crotty, 2021*; *Tan et al., 2021a*). Additionally, SARS-CoV-2-specific T cell responses have been shown to be present in most individuals 6 months after infection or vaccination and remain largely unaffected by emerging variants of concern, illustrating their importance in generating durable immune responses (*Chiuppesi et al., 2022*; *Choi et al., 2022*; *Gao et al., 2022*; *GeurtsvanKessel et al., 2022*; *Jung et al., 2022*; *Keeton et al., 2022*; *Liu et al., 2022*; *Redd et al., 2022*; *Tarke et al., 2022*).

Besides de novo SARS-CoV-2-specific T cell responses in infected individuals, SARS-CoV-2-specific T cells have also been identified in unexposed individuals (*Grifoni et al., 2020*; *Le Bert et al., 2020*; *Mateus et al., 2020*; *Nelde et al., 2021*; *Weiskopf et al., 2020*). This finding indicates that T cells which were initially primed against other pathogens are able to cross-recognize SARS-CoV-2 antigen. This phenomenon is called heterologous immunity and can often be explained by genomic sequence homology between pathogens. Highly homologous DNA sequences are translated into similar proteins which can be processed and presented as epitopes with high sequence similarity in human leukocyte antigen (HLA). For this reason, most research has focused on cross-reactive T cells that are potentially primed by other human coronaviruses (HCoVs) since they share around 30% amino acid sequence homology with SARS-CoV-2 (*Bacher et al., 2020*; *Braun et al., 2020*; *Johansson et al., 2021*; *Kundu et al., 2022*; *Le Bert et al., 2020*; *Loyal et al., 2021*; *Mateus et al., 2020*). However, it has been postulated that SARS-CoV-2-specific T cells in unexposed individuals could also conceivably be primed by other, non-HCoVs (*Le Bert et al., 2020*; *Peng et al., 2020*; *Stervbo et al., 2020*; *Tan et al., 2021b*). Furthermore, previous studies, although limited, have demonstrated the occurrence of cross-reactivity between two epitopes with relatively low sequence homology (*Bijen et al., 2018*; *Cameron et al., 2013*; *Clute et al., 2005*; *Cornberg et al., 2010*; *Riley et al., 2018*; *Su and Davis, 2013*). This form of heterologous immunity is poorly understood, and therefore, predicting such cross-reactivity remains a challenge (*Lee et al., 2020*).

Pre-pandemic SARS-CoV-2-specific T cells are reportedly present in a relatively high proportion of the population, independent of geographical location, indicating that a highly prevalent pathogen could be the initial trigger of these cross-reactive T cells (*Braun et al., 2020*; *Grifoni et al., 2020*; *Le Bert et al., 2020*; *Mateus et al., 2020*; *Meckiff et al., 2020*; *Nelde et al., 2021*; *Sekine et al., 2020*; *Weiskopf et al., 2020*). Furthermore, these cross-reactive T cells should be present in relatively high frequencies, as they are detectable in antigen-induced stimulation assays without additional amplification steps (*Braun et al., 2020*; *Grifoni et al., 2020*; *Le Bert et al., 2020*; *Sekine et al., 2020*; *Weiskopf et al., 2020*). Cytomegalovirus (CMV) is a highly prevalent pathogen and usually induces high T cell frequencies, making CMV a potential trigger for cross-reactive SARS-CoV-2-specific T cells (*Sylwester et al., 2005*; *Zuhair et al., 2019*). This is supported by the finding that SARS-CoV-2 cross-reactive CD8$^+$ T cells were increased in CMV-seropositive (CMV$^+$) donors and that previous CMV infection has been associated with severe COVID-19 (*Alanio et al., 2022*; *Jo et al., 2021*; *Weber et al., 2022*). Studies so far indicate that cross-reactive T cells can play a role in COVID-19 immunity, but whether they are protective or pathogenic is unclear (*Bacher et al., 2020*; *Kundu et al., 2022*). Taken together, we hypothesized that cross-reactive SARS-CoV-2-specific T cells might originate from the CMV-specific memory population.

In the present study, we aimed to identify SARS-CoV-2-specific cross-reactive CD4$^+$ and CD8$^+$ T cells in SARS-CoV-2-unexposed individuals. We found an increased presence of cross-reactive T cells in CMV$^+$ donors, and upon isolation and clonal expansion of the spike-reactive CD8$^+$ and membrane-reactive CD4$^+$ T cells, we confirmed that these T cells were reactive against both SARS-CoV-2 and CMV. Interestingly, isolated CD8$^+$ T cells recognizing a previously described CMV epitope IPSINVHHY presented by HLA-B*35:01 were cross-reactive with dissimilar SARS-CoV-2 spike peptide FVSNGTHWF presented by HLA-B*35:01, demonstrating that cross-reactivity does not solely depend on peptide sequence homology. The T cell receptor (TCR) isolated from these CD8$^+$ T cells was found in multiple donors showing that pre-pandemic spike-reactive CD8$^+$ T cells can be caused by a public CMV-specific TCR. Based on the reduced activation status compared to other SARS-CoV-2-specific T cells in severe COVID-19 patients, we hypothesize that these cross-reactive T cells are not important for clearing the virus at this late stage of the disease. However, these cross-reactive CD8$^+$ T cells were shown to reduce spreading of SARS-CoV-2 infection in vitro, and in two out of two CMV$^+$ severe COVID-19 patients, these cross-reactive T cells were detected. This indicates that early in infection at

the stage that no SARS-CoV-2-specific T cells are present yet, and these cross-reactive T cells may play a role in preventing SARS-CoV-2 infection or reducing the severity of COVID-19.

## Results

### SARS-CoV-2-specific T cell responses in SARS-CoV-2-unexposed PBMCs correlate with CMV seropositivity

To investigate whether SARS-CoV-2-specific CD4+ and CD8+ T cell responses in SARS-CoV-2-unexposed donors correlate with previous CMV infection, pre-pandemic cryopreserved PBMCs from CMV sero-positive (CMV+, N=28) and CMV seronegative (CMV−, N=39) healthy individuals were stimulated over-night using SARS-CoV-2 15-mer peptide pools. These pools included three spike peptide pools that together overlap the entire spike gene (S, S1, and S+), membrane (M), and nucleocapsid (N) antigens from SARS-CoV-2. To confirm that CMV+ individuals have CMV-specific T cells, reactivity against the most immunogenic CMV antigen, pp65, was also tested. Memory SARS-CoV-2-specific CD4+ T cells were characterized as CD154+CD137+, and memory SARS-CoV-2-specific CD8+ T cells were identified based on expression of CD137 and IFN-γ (*Figure 1A–B* and *Figure 1—figure supplement 1*). As expected, all CMV+ donors displayed a CD4+ and/or CD8+ T cell response upon stimulation with pp65 (*Figure 1C–E*). No marked increase of CD4+ T cell responses was observed after SARS-CoV-2 spike and nucleocapsid stimulation in the CMV+ group compared to CMV−. However, six donors in the CMV+ group displayed a CD4+ T cell response against the membrane peptide pool which was not observed in the CMV− group (*Figure 1C*). Furthermore, CD4+ T cell response against the membrane pool was accompanied by a CD4+ T cell response against pp65 (*Figure 1D*). In addition, CD8+ T cell responses were detected against spike peptides in two CMV+ donors which were not detected in CMV− donors (*Figure 1E*). Interestingly, donors with a high CD8+ T cell response against SARS-CoV-2 spike peptides additionally displayed strong reactivity against pp65 (*Figure 1F*). Taken together, these results show that SARS-CoV-2-unexposed CMV+, but not CMV−, individuals had detectable CD4+ T cell responses against membrane peptides and CD8+ T cells targeting spike peptides. These SARS-CoV-2 responses were accompanied by T cell responses against pp65 and thus may indicate that SARS-CoV-2 T cell responses in pre-pandemic samples potentially are memory T cells targeting pp65.

### Pre-pandemic SARS-CoV-2-specific CD4+ and CD8+ T cells recognize pp65 peptides from CMV

To confirm that pre-pandemic SARS-CoV-2-specific T cells are able to recognize peptides from pp65, these SARS-CoV-2-specific T cells were isolated and clonally expanded. SARS-CoV-2-unexposed (pre-pandemic cryopreserved) PBMCs from a CMV+ individual showing a CD4+ T cell response against SARS-CoV-2 membrane protein (donor UGT) were stimulated with the membrane peptide pool and single cell sorted based on CD137 upregulation (*Figure 2—figure supplement 1*). After clonal expansion, 20 out of 27 screened T cell clones produced IFN-γ when stimulated with membrane peptide pool compared to no peptide stimulation (data not shown). T cell clones 4UGT5, 4UGT8, and 4UGT17, all three expressing a different TCR, were used for further experiments (*Figure 2—figure supplement 2A*). As hypothesized, the T cell clones were reactive against both SARS-CoV-2 membrane antigen and CMV pp65 when loaded on HLA-matched EBV-lymphoblastoid cell lines (EBV-LCLs; *Figure 2A*). Interestingly, IFN-γ production by the T cell clones was significantly increased when stimulated with pp65 peptides compared to membrane peptides, indicating higher avidity for CMV compared to SARS-CoV-2 (*Figure 2B*). To identify which peptide in pp65 is recognized, reactivity of T cell clone 4UGT8 against a pp65 library was measured which resulted in recognition of three sub pools which contained peptide AGILARNLVPM (*Figure 2—figure supplement 2B–C*). HLA-mismatched EBV-LCLs were retrovirally transduced with HLA Class II molecules that were commonly shared between donors that had a detectable CD4+ T cell response against the membrane and pp65 peptide pool (*Figure 1D*). T cell clone 4UGT8 recognized both peptide pools and the AGI peptide only when presented in HLA-DRB3*02:02 (*Figure 2C* and *Figure 2—figure supplement 2D*). The SARS-CoV-2 membrane protein epitope recognized by these cross-reactive T cells remains unidentified as in vitro experiments and in silico prediction methods failed to identify the epitope. A similar approach was applied for CD8+ T cells in which T cell clones were generated after SARS-CoV-2 spike peptide pool stimulation of PBMCs from CMV+ donor UTT (*Figure 2—figure supplement 1*). The isolated CD8+

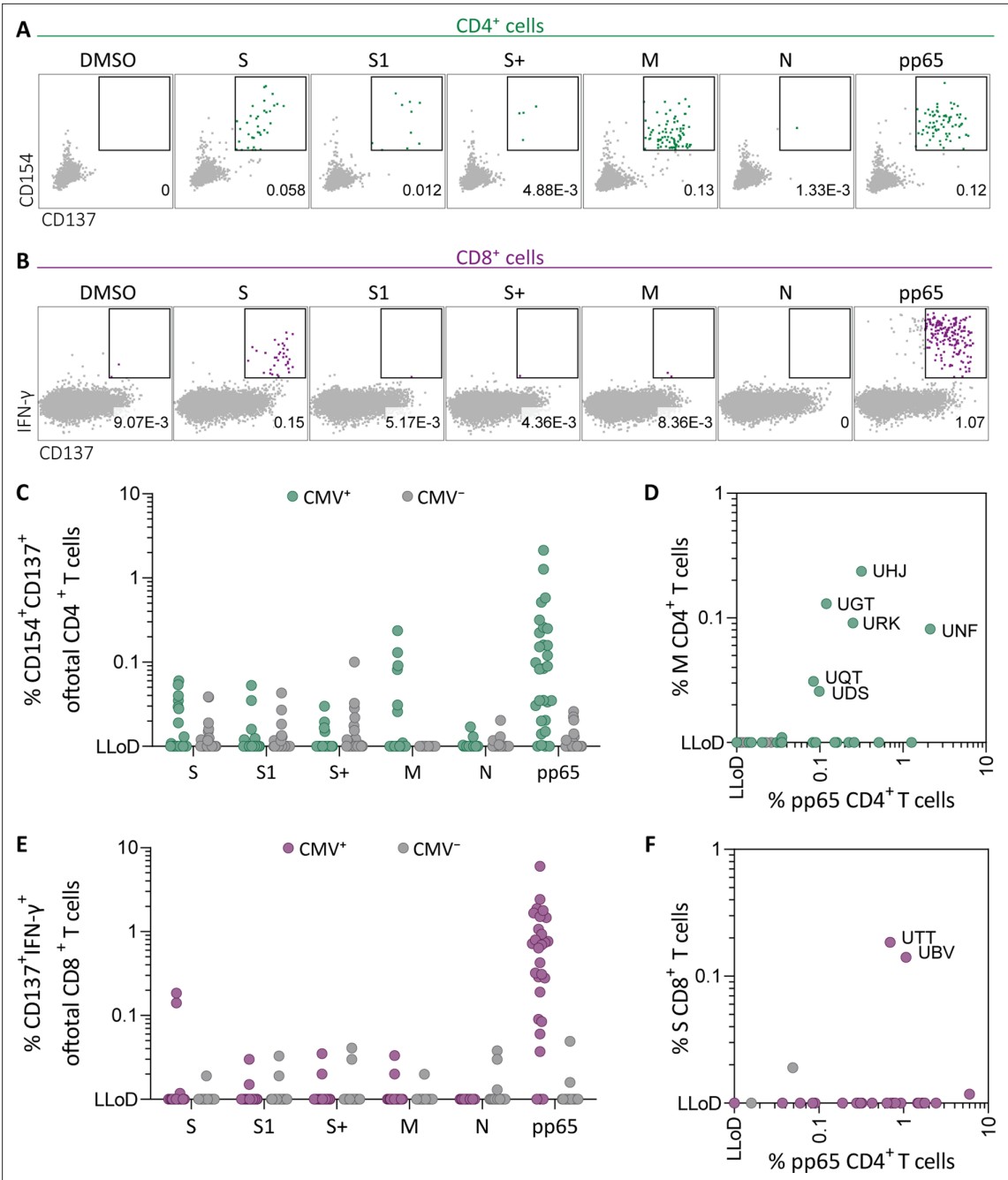

**Figure 1.** Ex vivo severe acute respiratory syndrome coronavirus 2 (SARS-CoV-2)-specific CD4+ and CD8+ T cell responses in cytomegalovirus (CMV)-positive and -negative unexposed donors. Pre-pandemic cryo-preserved PBMCs were stimulated using SARS-CoV-2 spike (S, S1, and S+), membrane (M), nucleocapsid (N), and CMV pp65 peptide pools or not stimulated (dimethylsulfoxide; DMSO). (**A**) A representative flow cytometry example of a CD4+ T cell response in a SARS-CoV-2-unexposed donor. Numbers in plot represent frequencies of CD137+CD154+ cells of total CD4+ T cells. (**B**) A representative flow cytometry example of a CD8+ T cell response in a SARS-CoV-2-unexposed donor. Numbers in plot represent frequencies of CD137+IFN-γ+ cells of total CD8+ T cells. (**C**) Scatter plot showing frequencies of CD137+CD154+ cells of total CD4+ T cells of CMV+ (green, N=28) and CMV– (gray, N=39) donors. (**D**) Frequencies of CD137+CD154+ cells of total CD4+ T cells in the membrane-stimulated condition (membrane response) plotted against pp65-stimulated condition (pp65 response). Three letter codes are anonymized codes of CMV+ (green) and CMV– (gray) donors. (**E**) Scatter plot showing frequencies of CD137+ IFN-γ+ cells of total CD8+ T cells of CMV+ (green, N=28) and CMV– (gray, N=39) donors. (**F**) Frequencies of CD137+IFN-γ+ cells of total CD8+ T cells in the spike-stimulated condition (spike response) plotted against pp65-stimulated condition (pp65 response).

The online version of this article includes the following source data and figure supplement(s) for figure 1:

*Figure 1 continued on next page*

*Figure 1 continued*

**Source data 1.** Source data containing the percentages underlying *Figure 1C–F*.

**Figure supplement 1.** Flow cytometry gating example for peptide stimulation assays.

T cell clones were screened for their reactivity with SARS-CoV-2 spike which showed that 23 out of the 28 T cell clones produced IFN-γ upon spike peptide pool stimulation (data not shown). TCR sequencing revealed that all 23T cell clones expressed the same TCR (*Figure 2—figure supplement 3A*). T cell clone 8UTT6 was selected for further testing and analyzed for its cross reactivity toward

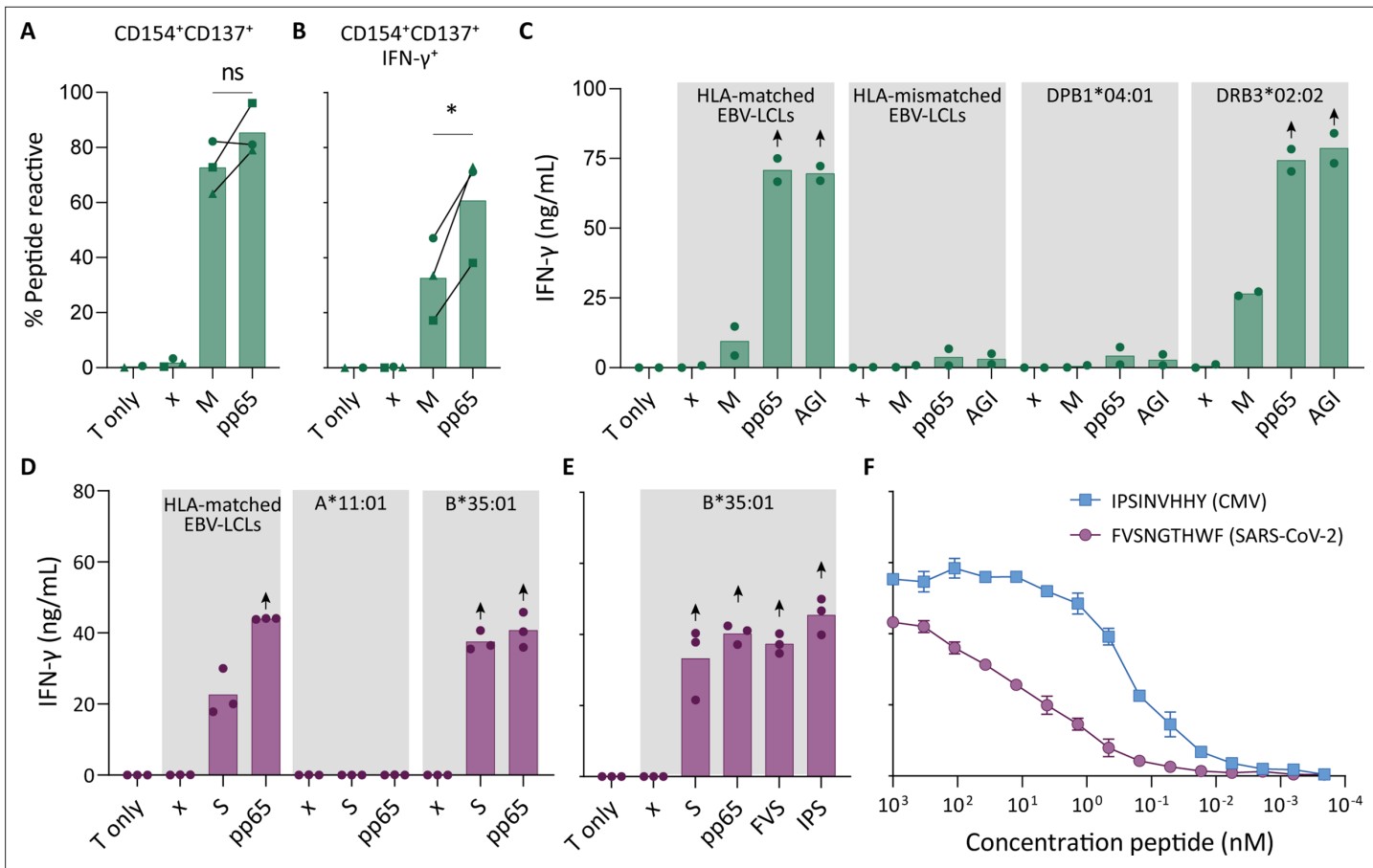

**Figure 2.** Recognition of severe acute respiratory syndrome coronavirus 2 (SARS-CoV-2) and cytomegalovirus (CMV) by pre-existing CD4+ and CD8+ T cells. Clonally expanded CD4+ T cells from donor UGT and CD8+ T cells from donor UTT were overnight co-cultured with peptide-pulsed stimulator cells. (**A–B**) Percentages of CD154+, CD137+, and/or IFN-γ+ cells of cross-reactive CD4+ T cell clones after overnight culture (T only) or after overnight co-culture with HLA-matched EBV-lymphoblastoid cell lines (EBV-LCLs) that were not peptide pulsed (x) or loaded with membrane (M) or pp65 peptide pool, measured by flow cytometry. Dots represent the mean of experimental repeats of 4UGT5 (square, one repeat), 4UGT8 (circles, four repeats), and 4UGT17 (triangle, two repeats). Significance was tested by a paired *t*-test. (**C**) Bar graphs showing ELISA measurement of secreted IFN-γ after co-culturing of a representative clone, 4UGT8 clone, with HLA-matched or HLA-mismatched EBV-LCLs. HLA-mismatched EBV-LCLs were retrovirally transduced with HLA class II molecule as depicted in figure. Stimulator cells were peptide-pulsed with membrane (M) peptide pool, pp65 peptide pool, or AGILARNLVPM (AGI) peptide. Data points are experimental duplicates. Black arrows indicate that values were above plateau value of the ELISA calibration curve. (**D–E**) Bar graphs showing ELISA measurement of secreted IFN-γ after co-culturing of a representative clone, 8UTT6 clone, with HLA-matched EBV-LCLs or K562s transduced with HLA-B*35:01 or HLA-A*11:01. Stimulator cells were peptide-pulsed with spike (S) peptide pool, pp65 peptide pool, IPSINVHHY (IPS) peptide, or FVSNGTHWF (FVS) peptide. Data points are technical triplicates. (**F**) Peptide titration of IPS peptide (blue) and FVS peptide (purple) in a co-culture assay with 8UTT6 clone.

The online version of this article includes the following figure supplement(s) for figure 2:

**Figure supplement 1.** Flow-activated cell sorting gating example for peptide stimulation assays.

**Figure supplement 2.** T cell receptor (TCR) sequence and pp65 peptide identification of 4UGT8 clone.

**Figure supplement 3.** T cell receptor (TCR) sequence and peptide identification of 8UTT6 clone.

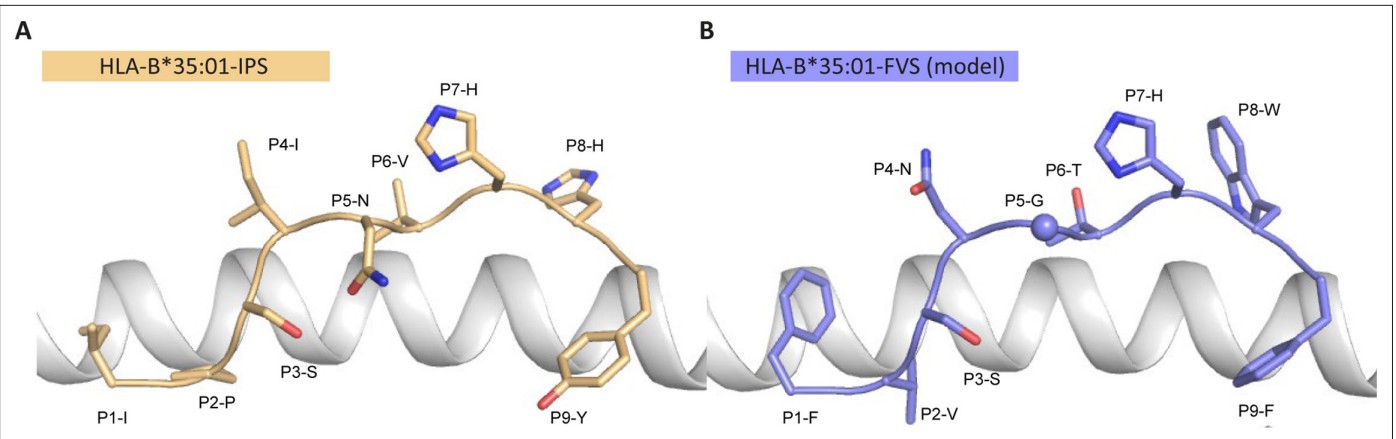

**Figure 3.** Model of the HLA-B*35:01-FVS structure. (**A**) Crystal structure of the HLA-B*35:01-IPS complex with the HLA in white cartoon and the IPS peptide in clear orange cartoon and stick. (**B**) Model of the HLA-B*35:01-FVS complex with the HLA in white cartoon and the FVS peptide in blue cartoon and stick. The sphere represents the Cα atom of the FVS peptide P5-G residue.

The online version of this article includes the following figure supplement(s) for figure 3:

**Figure supplement 1.** Structural overlay of HLA-B*35:01-IPF structure with the model of the HLA-B*35:01-FVS.

SARS-CoV-2 spike and CMV pp65 peptide pools. Additionally, the HLA restriction of T cell clone 8UTT6 was hypothesized to be HLA-B*35:01 as the unexposed donors with a CD8[+] T cell response against SARS-CoV-2 spike (UTT and UBV) both expressed HLA-B*35:01. The results confirmed that T cell clone 8UTT6 recognized spike as well as pp65 peptide pool presented by K562 cells transduced with HLA-B*35:01 but not transduced with HLA-A*11:01 (*Figure 2D*). To identify the spike epitope, reactivity of clone 8UTT6 against the 15-mer spike peptide library was measured. For the identification of the CMV epitope, an unbiased approach was performed using the nonamer combinatorial peptide library (CPL) assay. Recognition patterns were analyzed using netMHC 4.0 analysis for predicted binding to HLA-B*35:01, which revealed SARS-CoV-2 spike peptide FVSNGTHWF (FVS, $S_{1094–1103}$) and CMV pp65 IPSINVHHY (IPS, $pp65_{112-121}$) as the most likely epitopes (*Figure 2—figure supplement 3B–E*). The FVS and IPS peptides were indeed recognized by clone 8UTT6 (*Figure 2E*). Importantly, the IPS peptide was recognized with higher avidity compared to the FVS peptide by clone 8UTT6 (*Figure 2F*). Supporting these findings, the same TCRβ chain was already described and demonstrated to be specific for IPS in HLA-B*35:01 (*Klarenbeek et al., 2012*). Taken together, SARS-CoV-2 reactive CD4[+] and CD8[+] T cells in pre-pandemic samples cross reacted with CMV and SARS-CoV-2 peptides.

## Similarity at the C-terminal part of the peptides could drive T cell cross reactivity

To understand the molecular basis of T cell cross reactivity between dissimilar peptides FVS and IPS, we modeled the FVS structure based on the solved structure of the IPS peptide bound to HLA-B*35:01 (*Figure 3*; *Pellicci et al., 2014*). The two peptides share two residues (P3-S and P7-H) and have two similar residues (P6-T/V and P9-F/Y) based on similar biochemical properties and size. Residue substitutions from the IPS to FVS peptide were possible without major steric clashes with the HLA or peptide residues. The lack of secondary anchor residue at position 5 in the FVS peptide (P5-N/G) might change the conformation of the central part of the peptide that could be similar to the one observed in the spike-derived peptide IPF ($S_{896-904}$) in complex with HLA-B*35:01 (*Figure 3—figure supplement 1*; *Nguyen et al., 2021*). The primary anchor in the FVS peptide is P2-V and P9-F, both within the favored residues at those positions for HLA-B35-restricted peptide (*Escobar et al., 2008*). Overall, the FVS peptide might adopt a similar backbone conformation compared to the IPS peptide, which would place in both peptides a small hydrophobic residue at position 6 (P6-T/V), a histidine at position 7, and a residue with a large side-chain at position 8 (P8-W/H).

## IPS/FVS-specific cross-reactive CD8⁺ T cells are detectable in multiple individuals

To investigate the prevalence and phenotype of IPS/FVS cross-reactive T cells, HLA-B*35:01⁺ CMV⁺ healthy donors were screened for IPS/FVS-specific T cells using tetramers consisting of HLA-B*35:01-FVS (B*35/FVS-tetramer) and HLA-B*35:01-IPS (B*35/IPS-tetramer; *Figure 4—figure supplement 1*). Tetramer staining of PBMCs from donor UTT demonstrated that not all T cells that bound

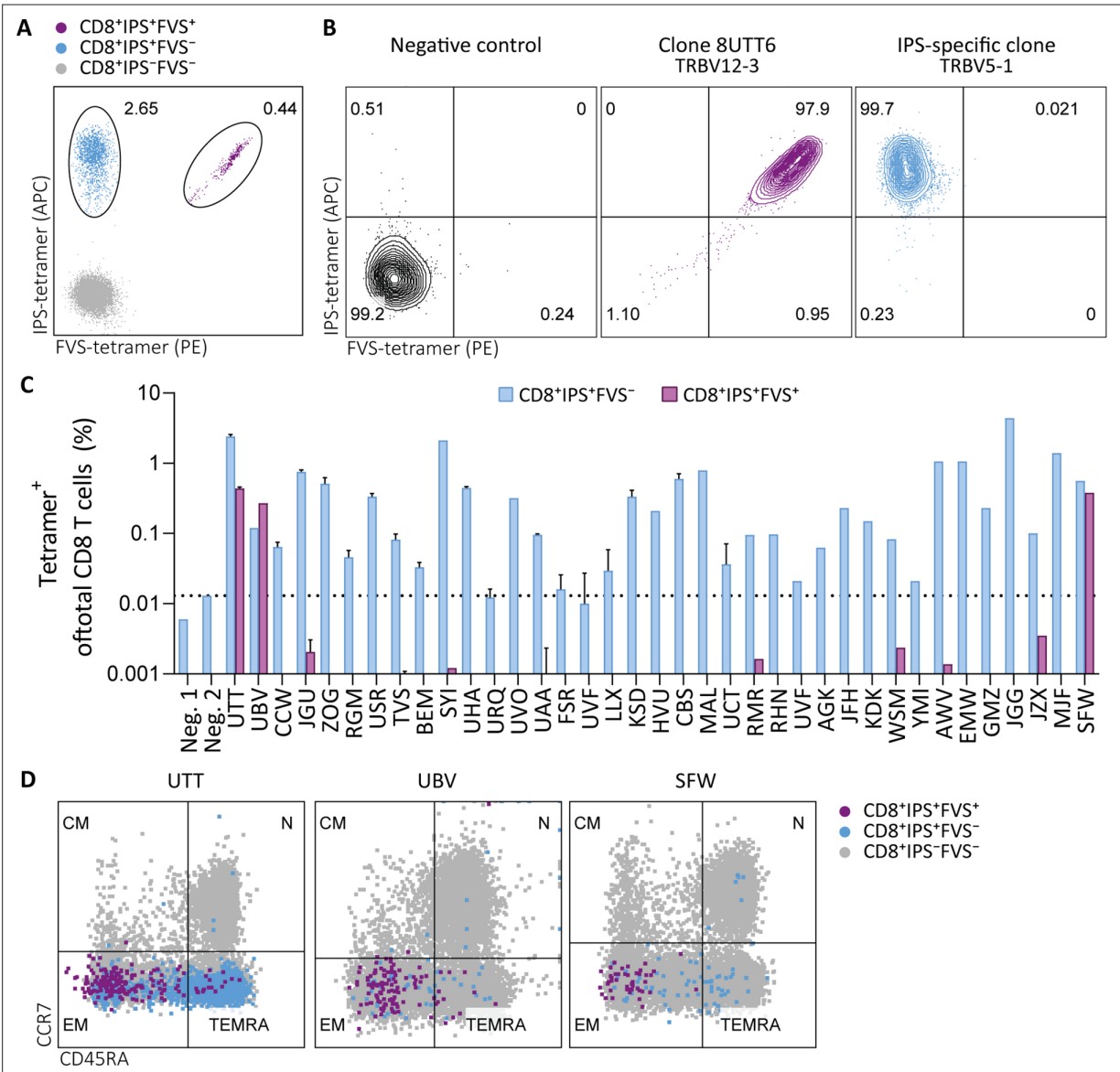

**Figure 4.** Tetramer detection of IPS/FVS-specific CD8⁺ T cells in CMV⁺ and HLA-B*35:01⁺ donors. Flow cytometry measurement of PBMCs or T cell clones that are binding to B*35/IPS-tetramer (blue), B*35/FVS-tetramer (purple), or to neither (gray). (**A**) Flow cytometry dot plot showing percentages of tetramer-binding cells of total CD8⁺ T cells in PBMCs from donor UTT. (**B**) Dot plot showing percentages of tetramer-binding of 8UTT6 clone and an IPS-specific clone with their international immunogenetics information system (IMGT) variable region of T cell receptor β-chain (TRBV) depicted. As a negative control (neg. ctrl.), a T cell clone recognizing a non-relevant peptide in HLA-B*35:01 was included. (**C**) Bar graph showing frequencies of tetramer-binding of total CD8⁺ from PBMCs of healthy CMV⁺ and HLA-B*35(:01)⁺ donors. Error bars represent SD of experimental duplicates. Dotted line represents background level which was based on HLA-B*35:01⁻ donors (neg.). (**D**) Dot plot showing expression of CCR7 and CD45RA by total CD8⁺ T cells and tetramer-binding T cells in PBMCs from UTT, UBV, and SFW. Quadrants separate differentiation subsets into naïve (N), central memory (CM), effector memory (EM), and terminally differentiated effector memory (TEMRA).

The online version of this article includes the following figure supplement(s) for figure 4:

**Figure supplement 1.** Flow cytometry gating example for tetramer staining assays.

to B*35/IPS-tetramer were able to bind to the B*35/FVS-tetramer as well. However, all T cells that bound to B*35/FVS-tetramer were also binding to the B*35/IPS-tetramer (*Figure 4A*). This observation indicates that IPS/FVS cross-reactivity is dictated by specific TCR sequences which were further supported by the lack of binding to B*35/FVS-tetramer by an IPS-specific T cell clone with a different TCR (*Figure 4B*). Screening of SARS-CoV-2-unexposed, CMV[+], and HLA-B*35(:01) donors (N=37) showed that nearly all CMV[+] donors had IPS-specific T cells with frequencies above background level, and interestingly, three of the analyzed donors (UTT, UBV, and SFW) presented with clearly detectable IPS/FVS-specific T cells (*Figure 4C*). Furthermore, IPS/FVS-specific T cells displayed an effector memory phenotype (CCR7[−]CD45RA[−]), confirming a memory repertoire origin and, interestingly, a less differentiated phenotype compared to IPS-specific T cells (*Figure 4D*). In summary, IPS/FVS cross reactivity is dependent on the TCR clonotype, and these cross-reactive T cells are detected in multiple donors.

## IPS/FVS cross reactivity is underpinned by a public TCR

To investigate whether the IPS/FVS-specific CD8[+] T cells found in multiple donors expressed a similar TCR, B*35/FVS-tetramer-binding T cells were isolated and the TCR α and β chains sequenced (*Figure 5—figure supplement 1*). Sequencing was performed for samples with clear detection of IPS/FVS-specific T cells (UTT, UBV, and SFW) and one donor with detectable, but below the limit of

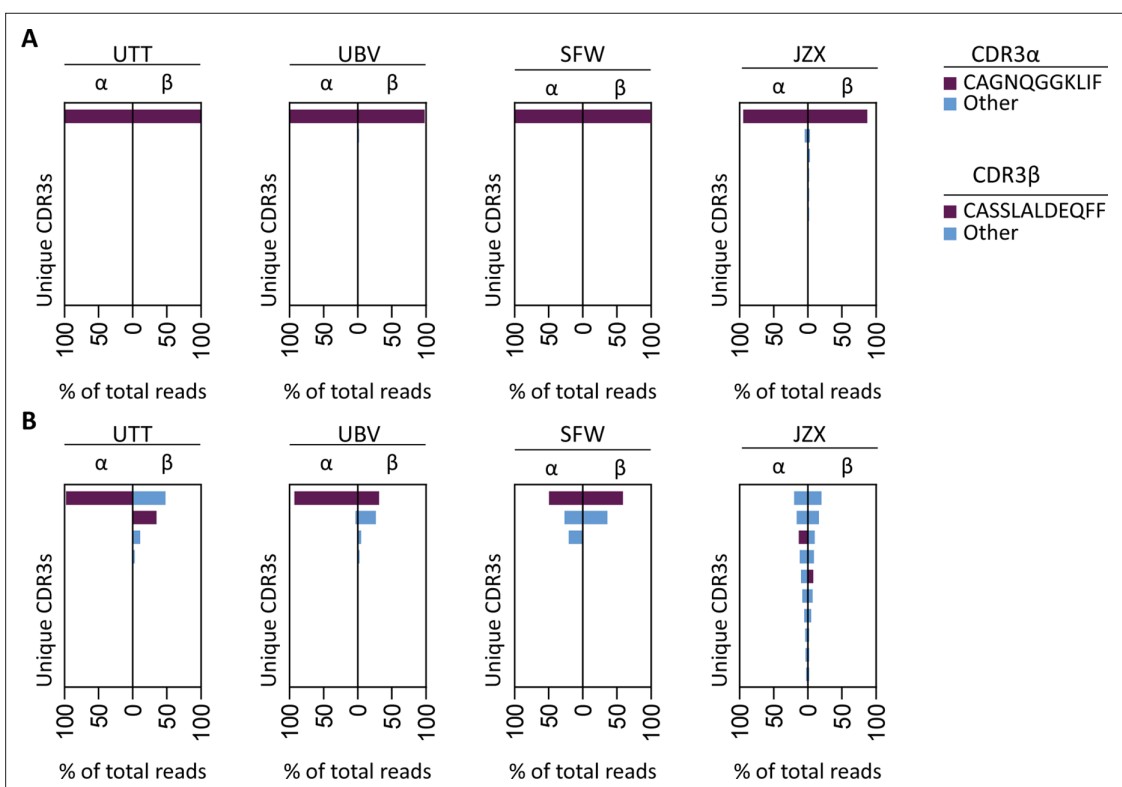

**Figure 5.** T cell receptor (TCR) sequencing of IPS/FVS-specific T cells. PBMCs from healthy CMV[+] and HLA-B*35:01[+] donors were sorted on B*35/IPS- or B*35/FVS-tetramer binding and directly sequenced for their TCR alpha and beta chain. Unique CDR3 sequences are depicted in two-sided bar graphs in which the left side shows abundance of CDR3 sequences from the TCR α-chain (CDR3α), and the right side shows abundance of CDR3 sequences from the TCR β-chain (CDR3β). Bar graphs are purple if the CDR3α has the CAGNQGGKLIF sequence, or the CDR3β has the CASSLALDEQFF sequence; all other found sequences are depicted in blue. CDR3s with less than 1% abundance were excluded from the analysis. (**A**) Two-sided bar graphs showing abundances of unique CDR3 sequences of samples sorted on binding to B*35/FVS-tetramer. (**B**) Two-sided bar graphs showing abundances of unique CDR3 sequences of samples sorted on binding to B*35/IPS-tetramer.

The online version of this article includes the following source data and figure supplement(s) for figure 5:

**Source data 1.** T cell receptor sequences of IPS/B*35 and FVS/B*35-specific CD8+ T cells.

**Figure supplement 1.** Flow activated cell sorting gating example.

**Figure supplement 2.** T cell receptor (TCR) sequencing of B*35/FVS-sorted samples.

accurate detection of B*35/FVS-tetramer⁺ T cells (JZX; *Figure 4C*). Interestingly, B*35/FVS-isolated T cells from all donors displayed amino acid identical dominant complementary-determining region 3 (CDR3) of the α-chain, CAGNQGGKLIF (CDR3α^CAGNQG), and β-chain, CASSLALDEQFF (CDR3β^CASSLA; *Figure 5A*). This observation thereby shows that IPS/FVS cross reactivity is caused by a public TCR. These identical CDR3s were not a result of sequencing artifact as nucleotide alignment revealed minor differences between samples (*Figure 5—figure supplement 2*). In addition to B*35/FVS-isolated T cells, T cells that bound B*35/IPS-tetramer were isolated and sequenced in parallel. Both CDR3α^CAGNQG and CDR3β^CASSLA were identified in all samples and shown to be among the most dominant TCRs. Remarkably, this was also observed in donor JZX which showed IPS/FVS-tetramer⁺ T cells below background level, indicating that in more than 3 out of 37 donors, this public TCR is present. (*Figure 5C*). Taken together, IPS/FVS-specific T cells express an identical TCR, found in multiple donors, indicating that public TCRs can exhibit cross-reactive properties.

## IPS/FVS cross-reactive CD8⁺ T cells are able to recognize SARS-CoV-2-infected cells but do not show an activated phenotype during acute disease

To investigate whether IPS/FVS-specific CD8⁺ T cells can play a role during SARS-CoV-2 infection, the function of IPS/FVS-specific T cells in an in vitro model and the activation state of these T cells during acute SARS-CoV-2 infection in severe COVID-19 patients was assessed. Firstly, the reactivity of IPS/FVS-specific T cells against K562 transduced with the spike gene was measured which showed that the T cells were able to recognize endogenously processed and presented peptide (*Figure 6A*). To investigate whether the IPS/FVS-specific T cells can recognize SARS-CoV-2-infected cells and thereby limit viral spread, Calu-3 airway epithelial cells were infected with live SARS-CoV-2 virus (wildtype) and incubated for 6 hr before co-culturing with CD8⁺ T cells. SARS-CoV-2 spike-specific CD8⁺ T cells from a SARS-CoV-2 vaccinated donor were able to reduce intracellular SARS-CoV-2 RNA copies at both 0.05 and 0.5 multiplicity of infection (MOI) 24 hr post infection (*Figure 6B–C*). Interestingly, IPS/FVS-specific CD8⁺ T cells were able to reduce SARS-CoV-2 intracellular RNA copies in Calu-3 cells infected with 0.05 MOI (MOI *Figure 6B*). Incubating with 10-fold more virus (0.5 MOI) resulted in no difference in RNA copies compared to the no T cell control (*Figure 6C*). To further investigate the function of IPS/FVS-specific CD8⁺ T cells ex vivo, the activation state of these T cells was evaluated during severe COVID-19 disease in two CMV⁺ HLA-B*35:01⁺ patients. The activation state was measured by expression of activation markers CD38 and HLA-DR as these markers are highly expressed on SARS-CoV-2-specific CD8⁺ T cells during severe COVID-19 (*Figure 6D*; *Sekine et al., 2020*). Interestingly, IPS/FVS-cross-reactive T cells were detected in two out of two CMV⁺ HLA-B*35:01⁺ patients suffering from severe COVID-19, whereas the cross-reactive T cells were detected in 3 out of 37 healthy CMV⁺ HLA-B*35:01⁺ donors (*Figures 4C and 6E*). The expression of CD38 and HLA-DR was lower compared to the SARS-CoV-2-specific CD8⁺ T cells and not considerably increased compared to IPS-specific T cells that were not cross-reactive with FVS (*Figure 6D–E*). These results indicate that IPS/FVS-specific CD8⁺ T cells recognize SARS-CoV-2-infected cells and are able to limit SARS-CoV-2 replication at low virus titers. However, IPS/FVS-specific T cells did not show an activated phenotype during acute severe SARS-CoV-2 infection.

## Discussion

SARS-CoV-2-specific T cells in pre-pandemic cryo-preserved samples have been reported in several studies. The majority of these studies describe T cell immunity against other HCoVs as the main source of these T cells (*Bacher et al., 2020*; *Braun et al., 2020*; *Johansson et al., 2021*; *Kundu et al., 2022*; *Le Bert et al., 2020*; *Loyal et al., 2021*; *Mateus et al., 2020*). However, some studies have postulated that pre-pandemic SARS-CoV-2-specific T cells could be derived from other sources (*Le Bert et al., 2020*; *Peng et al., 2020*; *Stervbo et al., 2020*; *Tan et al., 2021b*). Our findings demonstrate that CMV pp65-specific CD4⁺ T cells cross react with the membrane protein from SARS-CoV-2, and CMV pp65-specific CD8⁺ T cells are able to cross react with SARS-CoV-2 spike protein. The cross-reactive CD8⁺ T cells recognized known CMV epitope IPSINVHHY in HLA-B*35:01 and cross reacted with the SARS-CoV-2 epitope FVSNGTHWF in HLA-B*35:01. These IPS/FVS-specific CD8⁺ T cells were detected in multiple donors all expressing an identical TCR, indicating that cross-reactivity with

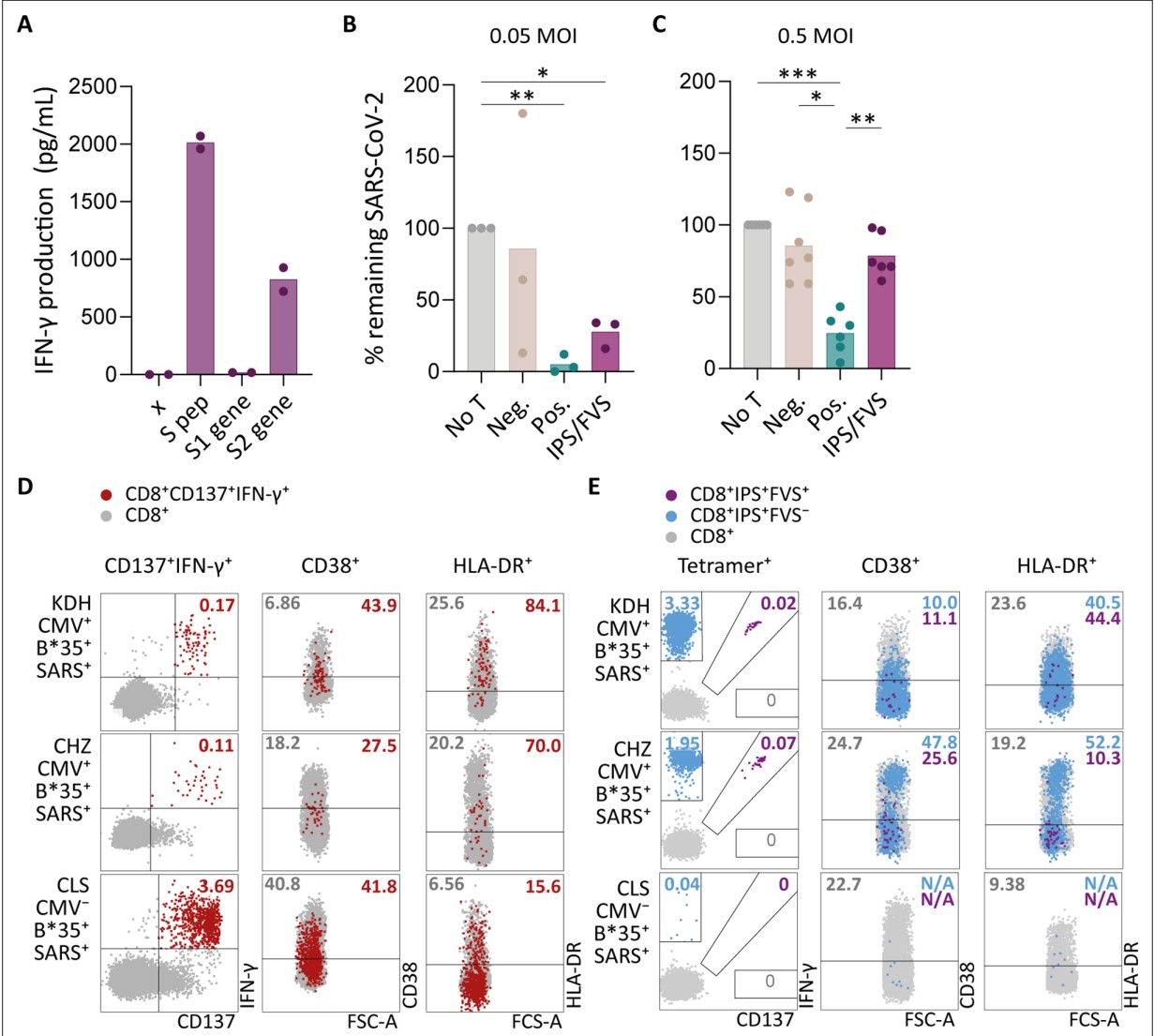

**Figure 6.** Ex vivo and in vitro evaluation of IPS/FVS-specific T cells. (**A**) IFN-γ release of IPS/FVS-specific CD8⁺ T cells after co-incubation with K562 that were untransduced (x), loaded with spike peptide pool (S pep), or transduced with nucleotide 1–2082 (S1 gene) or nucleotide 2052–3,822 (S2 gene) of the spike gene. (**B–C**) Calu-3 cells were transduced to express HLA-B*35:01 and infected with the wildtype severe acute respiratory syndrome coronavirus 2 (SARS-CoV-2) virus. 6 hr post infection (hpi), IPS/FVS-specific CD8⁺ T cells were added in a 10:1 effector to target ratio. SARS-CoV-2 spike-specific T cells, isolated from coronavirus disease 2019 (COVID-19) vaccinated individuals, that recognize VASQSIIAY presented in HLA-B*35:01 or YLQPRTFLL presented in HLA-A*02:01 functioned as a positive control (pos.) or negative control (neg.), respectively. Cells were harvested 24 hpi to measure intracellular viral RNA. Bar graphs show the means of percentage reduction in SARS-CoV-2 intracellular RNA copies compared to the no T cell condition (no T) as measured by reverse transcription quantitative PCR (RT-qPCR), at 24 hpi using a multiplicity of infection (MOI) of 0.05 or 0.5. One-way ANOVA was applied to test statistical differences between conditions and only comparisons with p<0.05 are shown. (**D–E**) Flow cytometry analysis of CD38 and HLA-DR expression on CD8⁺ T cells in PBMCs from severe COVID-19 patients who were CD137⁺IFN-γ⁺ after SARS-CoV-2 nucleocapsid peptide stimulation (red), only bound to B*35/IPS-tetramer (blue) or bound to both B*35/IPS- and B*35/FVS-tetramer (purple). All other CD8⁺ T cells are gray. Two patients were HLA-B*35:01⁺CMV⁺ (KDH and CHZ), and as a control, one patient was HLA-B*35:01⁺CMV⁻ (CLS). Detection of B*35/IPS- and B*35/FVS-specific T cells and expression of the activation markers were measured and compared within the same sample.

SARS-CoV-2 can be caused by a CMV-specific public TCR. Functional and phenotypic assessment of the IPS/FVS-specific CD8⁺ T cells indicated their capacity to reduce low concentrations of SARS-CoV-2 in vitro, but these cross-reactive T cells detected in two severe COVID-19 patients were not activated based on phenotypic characterization.

To our knowledge this is the first study to identify CMV-specific T cells that are cross reactive with SARS-CoV-2. The cross-reactive CD4⁺ T cells recognized CMV pp65 epitope AGILARNLVPM in HLA-DRB3*02:02 and were able to cross react with an as of yet unidentified, SARS-CoV-2 membrane epitope

in HLA-DRB3*02:02 (*Weber et al., 2022*). Previous studies have reported the presence of membrane-specific CD4[+] T cell responses in SARS-CoV-2-unexposed donors utilizing the same commercially available membrane peptide pool, yet these studies did not aim to identify the peptide-HLA restriction (*Bacher et al., 2020*; *Sekine et al., 2020*). AGI-specific CD4[+] T cells have been described to be cross reactive toward SARS-CoV-2 spike which is in contrast to our finding (*Weber et al., 2022*). The cross-reactive CD8[+] T cells recognize the CMV epitope IPSINVHHY and SARS-CoV-2 epitope FVSNGTHWF presented in HLA-B*35:01. IPS/FVS-specific T cells were possibly detected previously but never further investigated or characterized (*Shomuradova et al., 2020*; *Tarke et al., 2021*). Both cross-reactive CD4[+] and CD8[+] T cells displayed a higher avidity for the CMV epitope compared to the epitope derived from SARS-CoV-2. In contrast, other studies have reported an equal or even higher avidity for the SARS-CoV-2 epitope compared to the epitopes derived from the HCoV for which the T cells were hypothesized to be primed against (*Braun et al., 2020*; *Johansson et al., 2021*; *Lineburg et al., 2021*; *Mateus et al., 2020*). This appears to be contradictive since it has been shown that repeated exposure results in the selection of high avidity T cell clonotypes which are able to clear viral infection and protect against reinfection (*Abdel-Hakeem et al., 2017*; *Hombrink et al., 2013*; *Price et al., 2005*; *Schober et al., 2018*). Cross-reactive T cells would therefore most likely display a higher avidity for the source pathogen compared to the avidity for SARS-CoV-2, as reported in this study. This discrepancy could be caused by the fact that previous studies focused on other HCoVs since they share high sequence homology with SARS-CoV-2, thereby potentially missing the true source of these particular T cells (*Bacher et al., 2020*; *Braun et al., 2020*; *Johansson et al., 2021*; *Kundu et al., 2022*; *Le Bert et al., 2020*; *Loyal et al., 2021*; *Mateus et al., 2020*). Alternatively, samples frozen down during the pandemic were considered unexposed if the donors displayed neither SARS-CoV-2-specific antibodies nor a history of COVID-19-like symptoms (*Bacher et al., 2020*; *Braun et al., 2020*; *Lineburg et al., 2021*). However, SARS-CoV-2 infection does not necessarily lead to symptoms or a detectable antibody response (*Gao et al., 2021*; *Steiner et al., 2021*). The described reduced avidity for HCoV therefore could imply that these cross-reactive T cells were derived from the SARS-CoV-2-induced repertoire. Taken together, whereas cross-reactive T cells recognizing SARS-CoV-2 have been primarily described to be derived from other HCoVs, the contribution of these other HCoVs as initial primers of the T cell response may have been over-estimated due to experimental design. Further studies are required to identify other potential sources of cross reactivity with low sequence homology yet high prevalences such as CMV, EBV, influenza, or non-viral pathogens.

The identified cross-reactive CD8[+] T cells appeared to recognize CMV peptide IPSINVHHY and a dissimilar peptide FVSNGTHWF derived from SARS-CoV-2. Ex vivo detected heterologous CD8[+] T cell immunity against two pathogens caused by dissimilar epitopes presented in the same HLA is rarely reported (*Clute et al., 2005*; *Cornberg et al., 2010*). Nevertheless, ample studies have investigated the underlying mechanisms of such T cell-mediated cross reactivity. Heterologous immunity can be caused by the expression of a dual TCR which means that two TCR α- or β-chains are expressed simultaneously, resulting in two distinctive TCRs within one T cell (*Cusick et al., 2012*). However, here, we identified a single TCR in cross-reactive T cells excluding this hypothesis. Recognition of two distinct epitopes by a single TCR can be explained by shape similarity once the peptides are bound to the HLA molecule, and this shape similarity, or molecular mimicry can underpin T cell cross reactivity (*Macdonald et al., 2009*). Possible other underlying mechanisms are reduced footprint of the TCR with peptide (*Birnbaum et al., 2014*; *Cole et al., 2016*), an altered TCR-docking angle (*Adams et al., 2011*), or plasticity of the peptide-MHC complex (*Adams et al., 2011*; *Riley et al., 2018*) or TCR (*Piepenbrink et al., 2013*). Here, similarity between the IPS and FVS peptides in backbone conformation and the C-terminal part might underpin the T cell cross reactivity observed, as the majority of TCR docks preferentially toward the C-terminal of the peptide (*Szeto et al., 2020*). Solving the crystal structure of the IPS/FVS-TCR binding to HLA-B*35:01-FVS and -IPS would be necessary to provide insight in the binding properties of the public TCR.

IPS/FVS-specific CD8[+] T cells were able to reduce SARS-CoV-2 spread in vitro when exposed to a low virus concentration, which is supported by our finding that two out of two tested severe COVID-19 patients had clearly detectable IPS/FVS-specific CD8[+] T cells, while the prevalence in healthy donors was 3 out of 37. The presence of these cross-reactive memory T cells in circulation may be an advantage during initial SARS-CoV-2 infection as rapid T cell responses were associated with less severe COVID-19 (*Loyal et al., 2021*; *Sette and Crotty, 2021*; *Tan et al., 2021a*). However,

the cross-reactive CD8+ T cells were less efficient compared to SARS-CoV-2-specific, vaccination-primed T cells in limiting viral spread in vitro which can be explained by the reduced avidity of the cross-reactive T cells for the spike protein compared to CMV. This study also demonstrated that IPS/FVS-specific CD8+ T cells did not display the same degree of activation as observed for the SARS-CoV-2-specific T cells during severe COVID-19. Additionally, despite the presence of the cross-reactive CD8+ T cells, these individuals developed severe disease. These observations together indicate that IPS/FVS-specific CD8+ T cells might be able to reduce SARS-CoV-2 spread at initial infection but likely do not play a significant role in the pathogenesis of severe COVID-19. One limitation is that our study focused on circulating T cells, and we cannot exclude the possibility that cross-reactive CD8+ T cells present in lung tissue did display an activated phenotype. Another limitation of this study is the small severe COVID-19 cohort that was investigated, and literature describing the role of cross-reactive T cells is scarce (*Bacher et al., 2020*; *Kundu et al., 2022*). In summary, additional studies using larger cohorts are required to fully elucidate the potential role of cross-reactive CD8+ T cells in disease.

In conclusion, pre-pandemic SARS-CoV-2-specific T cells can derive from non-homologous pathogens such as CMV. This expands the potential origin of these pre-pandemic SARS-CoV-2-specific CD4+ and CD8+ T cell beyond other HCoVs. The cross-reactive CD8+ T cells were reactive toward dissimilar epitopes, and this cross reactivity was caused by a public TCR, which has been rarely observed so far. Our data points toward a role of the cross-reactive T cells in reducing SARS-CoV-2 viral load in the early stages of infection, prior to priming of SARS-CoV-2 specific T cells. Altogether, these results aid in further understanding heterologous T cell immunity beyond common cold coronaviruses and facilitate the investigation into the potential role of cross-reactive T cells in COVID-19.

# Methods

## Key resources table

| Reagent type (species) or resource | Designation | Source or reference | Identifiers | Additional information |
|---|---|---|---|---|
| Peptide, recombinant protein | Severe acute respiratory syndrome coronavirus 2 (SARS-CoV-2) Spike (S), 15-mers, 11aa overlapping peptide pool | Miltenyi | 130-126-701 | 1 µg/mL |
| Peptide, recombinant protein | SARS-CoV-2 Spike (S1), 15-mers, 11aa overlapping peptide pool | Miltenyi | 130-127-041 | 1 µg/mL |
| Peptide, recombinant protein | SARS-CoV-2 Spike (S+), 15-mers, 11aa overlapping peptide pool | Miltenyi | 130-127-312 | 1 µg/mL |
| Peptide, recombinant protein | SARS-CoV-2 Membrane (M), 15-mers, 11aa overlapping peptide pool | Miltenyi | 130-126-703 | 1 µg/mL |
| Peptide, recombinant protein | SARS-CoV-2 Nucleocapsid (N), 15-mers, 11aa overlapping peptide pool | Miltenyi | 130-126-699 | 1 µg/mL |
| Peptide, recombinant protein | Cytomegalovirus (CMV) pp65, 15-mers, 11aa overlapping peptide pool | JPT | Custom-made | 1 µg/mL |
| Peptide, recombinant protein | CMV pp65 peptide library, 15-mers, 11aa overlapping | JPT | Custom-made | 1 µg/mL |
| Peptide, recombinant protein | SARS-CoV-2 Spike peptide library, 15-mers, 11aa overlapping | SB Peptides | SB043 | 1 µg/mL |
| Peptide, recombinant protein | CMV, VFTWPPWQAGILARN | LUMC | Custom-made | 1 µg/mL |
| Peptide, recombinant protein | CMV, PPWQAGILARNLVPM | LUMC | Custom-made | 1 µg/mL |
| Peptide, recombinant protein | CMV, AGILARNLVPMVATV | LUMC | Custom-made | 1 µg/mL |
| Peptide, recombinant protein | CMV, ARNLVPMVATVQGQN | LUMC | Custom-made | 1 µg/mL |
| Peptide, recombinant protein | CMV, VPMVATVQGQNLKYQ | LUMC | Custom-made | 1 µg/mL |

*Continued on next page*

*Continued*

| Reagent type (species) or resource | Designation | Source or reference | Identifiers | Additional information |
|---|---|---|---|---|
| Peptide, recombinant protein | CMV, AQGDDDVWTSGSDSD | LUMC | Custom-made | 1 µg/mL |
| Peptide, recombinant protein | CMV, SSATACTSGVMTRGR | LUMC | Custom-made | 1 µg/mL |
| Peptide, recombinant protein | CMV, PKRRRHRQDALPGPC | LUMC | Custom-made | 1 µg/mL |
| Peptide, recombinant protein | SARS-CoV-2, FVSNGTHWF | LUMC | Custom-made | 1 µg/mL |
| Peptide, recombinant protein | CMV, IPSINVHHY | LUMC | Custom-made | 1 µg/mL |
| Antibody | Rat monoclonal anti-human CCR7 (BV711) | BD Biosciences | Cat.#563712 RRID:AB_2738386 | FC (1:100) |
| Antibody | Mouse monoclonal anti-human CD137 (APC) | BD Biosciences | Cat.#550890 RRID:AB_398477 | FC (1:75) |
| Antibody | Mouse monoclonal anti-human CD14 (FITC) | BD Biosciences | Cat.#555397 RRID:AB_395798 | FC (1:100) |
| Antibody | Mouse monoclonal anti-human CD154 (Pacific Blue) | Biolegend | Cat.#310820 RRID:AB_830699 | FC (1:300) |
| Antibody | Mouse monoclonal anti-human CD19 (FITC) | BD Biosciences | Cat.#555412 RRID:AB_395812 | FC (1:100) |
| Antibody | Mouse monoclonal anti-human CD4 (PE-Cy7) | Beckham Coulter | Cat.#737660 RRID:AB_2922769 | FC (1:300) |
| Antibody | Mouse monoclonal anti-human CD4 (FITC) | BD Biosciences | Cat.#555346 RRID:AB_395751 | FC (1:30) |
| Antibody | Mouse monoclonal anti-human CD4 (BV510) | BD Biosciences | Cat.#562970 RRID:AB_2744424 | FC (1:300) |
| Antibody | Mouse monoclonal anti-human CD45RA (PE-Texas-Red) | Invitrogen | Cat.#MHCD45RA17 RRID:AB_10372222 | FC (1:200) |
| Antibody | Mouse monoclonal anti-human CD8 (APC-H7) | BD Biosciences | Cat.#560179 RRID:AB_1645481 | FC (1:100) |
| Antibody | Mouse monoclonal anti-human CD8 (PE-Cy7) | BD Biosciences | Cat.#557746 RRID:AB_396852 | FC (1:320) |
| Antibody | Mouse monoclonal anti-human CD8 (Pacific Blue) | BD Biosciences | Cat.#558207 RRID:AB_397058 | FC (1:500) |
| Antibody | Mouse monoclonal anti-human IFN-γ (Alexa-Fluor 700) | Sony | Cat.#3112600 RRID:AB_2922770 | FC (1:120) |
| Antibody | Mouse monoclonal anti-human IFN-γ (BV711) | BD Biosciences | Cat.#564039 RRID:AB_2738557 | FC (1:300) |
| Antibody | Mouse monoclonal anti-human HLA-DR (Alexa-Fluor 700) | BD Biosciences | Cat.#560743 RRID:AB_1727526 | FC (1:150) |
| Antibody | Mouse monoclonal anti-human CD38 (BV605) | BD Biosciences | Cat.#740401 RRID:AB_2740131 | FC (1:120) |
| Antibody | Rat monoclonal anti-mouse CD19 (Mouse) | Biolegend | Cat.#557399 RRID:AB_396682 | FC (1:250) |
| Other | Zombie-Red | Biolegend | Cat.#423109 | FC (1:1000) |
| Other | Zombie-Aqua | BD Biosciences | Cat.#423101 | FC (1:1000) |
| Other | Brilliant Violet Staining Buffer Plus | Beckham Coulter | Cat.#566385 | FC (1:10) |
| Cell line (*Homo sapiens*) | K-562 | ATCC | CCL-342 | |
| Cell line (*Homo sapiens*) | Calu-3 | ATCC | HTB-55 | |
| Biological sample (*Homo sapiens*) | PBMCs from 67 healthy donors | LUMC Biobank | | Cryo-preserved before May 2019 |
| Biological sample (*Homo sapiens*) | PBMCs from critical COVID-19 patient (KDH) | LUMC BEAT-COVID consortium | Clinical trial #: NL8589 | Male, 61 years, 31 days ICU |
| Biological sample (*Homo sapiens*) | PBMCs from critical COVID-19 patient (CHZ) | LUMC BEAT-COVID consortium | Clinical trial #: NL8589 | Male, 76 years, 40 days ICU |

| Reagent type (species) or resource | Designation | Source or reference | Identifiers | Additional information |
|---|---|---|---|---|
| Biological sample (*Homo sapiens*) | PBMCs from critical COVID-19 patient (CLS) | LUMC BEAT-COVID consortium | Clinical trial #: NL8589 | Male, 71 years, 107 days ICU |

## Study samples and cell lines

Bio-banked PBMCs were cryopreserved after informed consent from the respective donors in accordance with the declaration of Helsinki. The samples from COVID-19 patients were part of a trial (NL8589) registered in the Dutch Trial Registry and approved by Medical Ethical Committee Leiden-Den Haag-Delft (NL73740.058.20). All three patients suffered from critical COVID-19 as categorized according to WHO guidelines (WHO ref#: WHO/2019-nCoV/clinical/2020.4; see *Supplementary file 1* for patient details). Bio-banked PBMCs from CMV-seropositive (N=28) and CMV-seronegative (N=39) donors that were frozen down before May 2019 were randomly selected to assure that the samples are SARS-CoV-2 naïve and represent the European population (*Supplementary file 2*). Prior to cryopreservation, PBMCs were isolated from fresh whole blood using Ficoll-Isopaque. PBMCs were thawed in culture medium consisting of Iscove Modified Dulbecco Medium (IMDM; Lonza, Basel, Switzerland) supplemented with 10% heat-inactivated fetal bovine serum (FBS; Sigma-Aldrich, Saint Louis, MO, USA), 2.7 mM L-glutamine (Lonza), 100 U/mL penicillin (Lonza), and 100 µg/mL streptomycin (Lonza; 1% p/s), and subsequently treated with 1.33 mg/mL DNAse to minimize cell clumping. K562 cells (*CCL-243;* American Type Culture Collection [ATCC]) and Calu-3 lung carcinoma cells (*HTB-55;* ATCC) were regularly checked for the presence of mycoplasma. K562s were regularly checked to ensure (lack of) HLA expression, and calu-3 cells were authenticated by STR sequencing.

## Intracellular cytokine staining assay

Thawed PBMCs were stimulated in culture medium supplemented with 1 µg/mL SARS-CoV-2 peptides pools covering the entire spike (Miltenyi, Keulen, Germany), membrane (Miltenyi), or nucleocapsid (Miltenyi) proteins for 1 hr at 37°C + 5% $CO_2$. The peptides of the spike gene were by the manufacturer divided over an 'S,' 'S1,' and 'S+' pool, wherein 'S' covers the most immunogenic parts of the gene, 'S1' mostly covers S1 domain, and 'S+' mostly covers S2 domain. An additional peptide pool containing 11 amino acid overlapping 15-mer peptides covering the pp65 antigen from CMV (JPT Peptide Technologies) was included (see *Supplementary file 3* for peptide details). After 1 hr stimulation, 5 µg/mL Brefeldin A (Sigma-Aldrich) was added, and the samples were incubated for an additional 15 hr at 37°C + 5% $CO_2$. The samples were subsequently stained with the viability dye Zombie-Red (Biolegend, San Diego, CA, USA) for 25 min at room temperature (RT) after which the cells were washed in PBS containing 0.8 mg/mL albumin (fluorescent activated cell sorting [FACS] buffer) and stained with antibodies against CD4 and CD8 in FACS buffer for 30 min at 4°C. Cells were washed in PBS and fixed in 1% paraformaldehyde for 8 min at RT followed by a wash and a permeabilization step for 30 min at 4°C in FACS buffer supplemented with 1% p/s and 0.1% saponin (permeabilization buffer). After permeabilization, the cells were stained using an antibody cocktail directed against CD14, CD19, CD137, CD154, and IFN-γ in permeabilization buffer (see *Supplementary file 4* for antibody details) for 30 min at 4°C. After staining, the samples were washed, resuspended in permeabilization buffer, and measured on a 3-laser aurora (Cytek Biosciences, Fremont, CA, USA).

## Isolation of SARS-CoV-2-specific T cells

Thawed PBMCs were stimulated for 16 hr at 37°C + 5% $CO_2$ using 1 µg/mL of spike (Miltenyi) or membrane (Miltenyi) peptide pool in culture medium (see *Supplementary file 3* for peptide details). After stimulation, the cells were washed and stained with antibodies directed against CD4, CD8, and CD137 in phenol-red free IMDM (Gibco, Waltham, MA, USA) containing 2% FBS (Sigma-Aldrich), 1% p/s (Lonza; sort medium; see *Supplementary file 4* for antibody details) for 30 min at 4°C. The cells were subsequently washed and resuspended in sort medium. CD4$^+$ or CD8$^+$ and CD137$^+$ cells were single-cell sorted using an Aria III cell sorter (BD Biosciences, Franklin Lakes, NJ, USA) into a 96-well round-bottom plate containing 1×10$^5$ 35-Gy-irradiated PBMCs, 50-Gy-irradiated EBV-LCL-JYs, and 0.8 µg/mL phytohemagglutinin (PHA) (Thermo Fisher, Waltham, MA, USA) in 100 µL T cell medium (TCM) consisting of IMDM (Lonza) supplemented with 2.7 mM L-glutamine (Lonza), 100 U/mL penicillin (Lonza), 100 µg/mL streptomycin (Lonza), 5% FBS (Sigma-Aldrich), 5% human serum (Sanquin,

Amsterdam, The Netherlands), and 100 IU/mL recombinant human IL-2 (Novartis, Basel, Switzerland). Sorted T cells were clonally expanded to generate T cell clones. T cell clones were restimulated between day 14 and day 20 post stimulation using PHA, PBMCs, and EBV-LCL-JYs as described above and used for assays between day 7 and day 20 post stimulation.

## Co-culture assays

To test peptide and HLA restriction, T cell clones were washed and co-cultured with stimulator cells in a 1:6 effector to stimulator ratio. Stimulator cells consisted of either autologous or HLA-matched EBV-LCLs or retrovirally transduced K562s. K562 were transduced with a pZLRS or MP71 vector containing a HLA gene of interested linked to a marker gene, and transduction was performed as previously described (*Jahn et al., 2015*). Cells were enriched for marker gene expression using magnetic activated cell sorting (MACS; Miltenyi) or FACS on an Aria III cell sorter (BD Biosciences). Stimulator cells were loaded with peptides through pre-incubation for 30 min at 37°C with 0.01–1 µM peptide (*Supplementary file 3* for peptide details). To identify the pp65 epitope of the CD4$^+$ T cell clones, a co-culture assay was performed using a pp65 peptide library. The pp65 library consisted of 15-mere peptides with 11 amino acid overlap, spanning the whole pp65 gene. The peptides are divided into matrix pools with horizontal and vertical sub pools so that each pool has an unique peptide combination, and each peptide is in one horizontal and one vertical sub pool. To identify the HLA-restriction of the CD4$^+$ T cell clones, the peptides were not washed away during the co-culture incubation period, and HLA class II was knocked out in the T cell clones as previously described (*Morton et al., 2020*). However, the protocol was adapted to knock-out Class II Major histocompatibility complex transactivator (CIITA) by designing two reverse guide RNAs: 5'-AGTCGCTCACTGGTCCCACTAGG-3' and 5'-CCGTGGACAGTGAATCCACTGGG-3' (Integrated DNA technologies Inc, Coralville, IA, USA). Co-culture assays were incubated overnight, and secreted IFN-γ was measured as an indicator of T cell activity by ELISA (Diaclone, Besançon, France) as described by the manufacturer.

To identify the peptide recognition signature of the CD8 T cell clones, a co-culture assay was performed using a nonamer CPL (*Bijen et al., 2018*). The 9-mer CPL scan contains 180 peptide pools with each pool consisting of a mixture of peptides with one naturally-occurring amino acid fixed at one position (*Wooldridge et al., 2010*). Co-culture assay was performed as described above with small changes; $2\times10^4$ K562 transduced with HLA-B*35:01 was pre-incubated with 100 µM CPL peptides for 1 hr at 37°C before $5\times10^3$ T cell clones were added. After overnight incubation, secreted IFN-γ was measured by an IFN-γ-ELISA (Diaclone), and results were analyzed using WSBC PI CPL for viruses (*Szomolay et al., 2016*; *Wooldridge, 2013*). Identified peptides following peptide libraries or CPL were analyzed for predicted binding to HLA-B*35:01 using netMHC 4.0 (*Andreatta and Nielsen, 2016*). Alternatively, peptide recognition by T cell clones was measured using intracellular cytokine staining (ICS) assay as described above.

## Peptide-HLA modeling

The binding of FVS in HLA-B*35:01 was modeled based on the solved crystal structure of the HLA-B*35:01-IPS (*Pellicci et al., 2014*). Each residue of the IPS peptide was mutated to their corresponding residues in the FVS peptide using the mutagenesis wizard in *PyMOL, 2015*. The residues were mutated into the most favorable rotamer to avoid steric clashes. No major steric clashes with the peptide or HLA were observed.

## Tetramer staining

$1–2\times10^6$ PBMCs or $5\times10^4$ T cell clones were incubated with in-house generated, PE- or APC-conjugated tetramers for 30 min at RT (*Hombrink et al., 2013*). After tetramer incubation, the cells were washed and incubated with an antibody mix targeting CD4, CD8, CD45RA, CCR7, CD38, and/or HLA-DR. After incubation, cells were washed and resuspended in FACS buffer and immediately measured on a 3-laser Aurora (Cytek Biosciences).

## TCR sequencing

PBMCs were thawed and $10–50\times10^6$ cells directly stained with PE-conjugated HLA-B*35:01-FVS or HLA-B*35:01-IPS tetramers. Tetramers were labeled to beads using anti-PE MicroBeads (Miltenyi) and enriched through magnetic-activated cell sorting (Miltenyi). The tetramer-enriched cells were

washed and incubated with an antibody cocktail targeting CD4 and CD8 (see *Supplementary file 4* for antibody details) in sort medium. Stained samples were washed in sort medium and bulk-sorted on an Aria III cell sorter (BD Biosciences; see *Figure 5—figure supplement 1* for a gating example). RNA isolation and TCR sequencing were performed as previously described (*Roukens et al., 2022*). In short, cells were directly collected in lysis buffer for RNA isolation using the ReliaPrep RNA cell Miniprep system (Promega, Madison, WI, USA). The total RNA yield of each sample was converted to cDNA using a template-switch oligo primer (Eurogentec, Seraing, Belgium), RNAsin (Promega), and SMARTScribe reverse transcriptase (Takara Bio, Kusatsu, Japan; *Koning et al., 2017*). cDNA was pre-amplified via an IS region in the Oligo dT primer prior to barcoding on samples containing cDNA from 500 or fewer cells (*Picelli et al., 2013*). Barcoded TCR PCR product was generated in two rounds of PCR: in the first PCR reaction, *TRA* and *TRB* product was generated in separate PCR reactions using Phusion Flash (Thermo Fisher Scientific), Smartseq2modified PCR primer (Eurogentec), and TRAC or TRBC1/2-specific primers (Eurogentec; see *Supplementary file 5* for primer list). The PCR product was then purified using the Wizard SV 96 PCR Clean-Up System (Promega) and barcoded in a second PCR using two-sided six-nucleotide barcoded primers to discriminate between TCRs of different T cell populations. PCR products of different T cell populations were pooled, after which TCR sequences were identified by NovaSeq (GenomeScan, Leiden, The Netherlands).

## SARS-CoV-2 infection assay

Calu-3 lung carcinoma cells (*HTB-55*; ATCC) were cultured in Eagle's minimum essential medium (Lonza), supplemented with 9% fetal calf serum (CapriCorn Scientific, USA), 1% NEAA (Sigma-Aldrich), 2 mM L-glutamine (Sigma-Aldrich), 1 mM sodium pyruvate (Sigma-Aldrich), and 100 U/ml of penicillin/streptomycin (P/S; Sigma-Aldrich). Calu-3 cells were retrovirally transduced with a pLZRS vector containing the HLA-B*35:01 molecule linked via an internal ribosome entry site sequence to mouse CD19, and transduction was performed as previously described (*Jahn et al., 2015*). Mouse CD19 was used as a marker gene to enrich for successfully transduced cells by adding antibodies directed against mouse CD19 and enriching for stained cells by MACS (Miltenyi) followed by FACS on an Aria III cell sorter (BD Biosciences; see *Supplementary file 4* for antibody details). For the infection assay, Calu-3 cells were seeded in 96-well cell culture plates at a density of $3 \times 10^4$ cells per well in 100 µL culture medium. Infections were done with clinical isolate SARS-CoV-2/Leiden-0008, which was isolated from a nasopharyngeal sample collected at the LUMC during the first wave of the Corona pandemic in March 2020 (GenBack: MT705206.1). Cells were infected with SARS-CoV-2 at a MOI of 0.05 or 0.5 in 50 µL infection medium. After 1.5 hr, cells were washed three times with medium, and 100 µL of medium was added. At 6 hr post infection (hpi), medium was removed again, and 100 µL of TCM with $3 \times 10^5$ T cells per well was added. At 24 hpi, cells were harvested to collect intracellular RNA by lysing the cells in 100 µL GITC reagent (3 M GITC, 2% sarkosyl, 20 mM Tris, and 20 mM EDTA) per well. Intracellular RNA was isolated using magnetic beads, and viral RNA was quantified by internally controlled multiplex TaqMan RT-quantitative PCR as described previously (*Salgado-Benvindo et al., 2020*).

## Statistics

Flow cytometry data was unmixed using Spectroflo (Cytek Biosciences) and analyzed using FlowJo v10.7.1. (BD Biosciences) to set gates on the samples based on the DMSO negative control in ICS assays or adapted to positive control for tetramer staining (see *Figure 1—figure supplement 1*, *Figure 2—figure supplement 1*, *Figure 4—figure supplement 1* and *Figure 5—figure supplement 1* for a gating example). Samples were excluded from the analysis if less than 10,000 events in CD4[+] or CD8[+] gate were measured or if after further testing they appeared not to be αβ T cells. For the SARS-CoV-2 infection assays, experiments were excluded from the analysis if the positive control had higher SARS-CoV-2 intracellular RNA copies compared to no T cell condition. Statistical analysis and generation of figures were conducted using GraphPad Prism 9.0.1 (GraphPad Software). Data was tested for significance using an one-way ANOVA with p-values below 0.05 considered as significant. p-Values are categorized in the figures as: ns = not significant; *p<0.05; **p<0.01; or ***p<0.001.

TCR sequence data was analysed using MiXCR software (v3.0.13) to determine the Vα and Vβ family and CDR3 regions using annotation of the IMGT library (http://www.imgt.org; v6) (*Bolotin*

*et al., 2015*). CDR3 regions were analysed in RStudio and CDR3 sequences that were non-functional or had ≤50 reads that were excluded from the analysis.

## Acknowledgements

The authors like to thank Joost M Lambooij for critically editing the manuscript. Flow cytometry was performed at the Flow cytometry Core Facility (FCF) of Leiden University Medical Center (LUMC) in Leiden, Netherlands (https://www.lumc.nl/research/facilities/fcf).

This study was financially supported by Health ~Holland (#LSHM19088) and Australian National Health and Medical Research Council (NHMRC), S.G. is supported by an NHMRC Senior Research Fellowship (#1159272).

## Additional information

### Funding

| Funder | Grant reference number | Author |
|---|---|---|
| Health~Holland | LSHM19088 | Mirjam HM Heemskerk |
| National Health and Medical Research Council | 1159272 | Stephanie Gras |
| Medical Research Council | | Stephanie Gras |

The funders had no role in study design, data collection and interpretation, or the decision to submit the work for publication.

### Author contributions

Cilia R Pothast, Formal analysis, Investigation, Visualization, Methodology, Writing – original draft; Romy C Dijkland, Formal analysis, Investigation; Melissa Thaler, Renate S Hagedoorn, Formal analysis, Investigation, Methodology; Michel GD Kester, Resources, Investigation, Methodology; Anne K Wouters, Investigation, Methodology; Pieter S Hiemstra, Methodology; Martijn J van Hemert, Supervision, Methodology; Stephanie Gras, Investigation, Visualization, Methodology, Writing – original draft; JH Frederik Falkenburg, Conceptualization, Supervision, Writing – review and editing; Mirjam HM Heemskerk, Conceptualization, Supervision, Funding acquisition, Methodology, Project administration, Writing – review and editing

### Author ORCIDs

Cilia R Pothast http://orcid.org/0000-0002-9761-3123
Romy C Dijkland http://orcid.org/0000-0002-3479-9303
Melissa Thaler http://orcid.org/0000-0002-5590-4918
Michel GD Kester http://orcid.org/0000-0002-8960-7348
Martijn J van Hemert http://orcid.org/0000-0002-2617-9243
Mirjam HM Heemskerk http://orcid.org/0000-0001-6320-9133

### Ethics

Clinical trial registration NL8589 (Dutch Trial Registry).
Human subjects: Bio-banked PBMCs were cryopreserved after informed consent from the respective donors, in accordance with the declaration of Helsinki. The samples from COVID-19 patients were part of a trial (NL8589) registered in the Dutch Trial Registry and approved by Medical Ethical Committee Leiden-Den Haag-Delft (NL73740.058.20).

### Decision letter and Author response

Decision letter https://doi.org/10.7554/eLife.82050.sa1
Author response https://doi.org/10.7554/eLife.82050.sa2

## Additional files

### Supplementary files
- MDAR checklist
- Supplementary file 1. COVID-19 patient characteristics.
- Supplementary file 2. Cohort characteristics of pre-pandemic samples.
- Supplementary file 3. List of peptides.
- Supplementary file 4. List of antibodies and reagents used for flow cytometry.
- Supplementary file 5. List of primer sequences used for TCR sequencing.

### Data availability
Figure 1 - Source data 1 contains percentages underlying figure 1C-F. Figure 4 - Source data 1 contains the sequence data used to generate figures and the data have been deposited in SRA (NCBI) database under BioProjectID PRJNA891934.

The following dataset was generated:

| Author(s) | Year | Dataset title | Dataset URL | Database and Identifier |
|---|---|---|---|---|
| Pothast CR, Hagedoorn RS, Heemskerk MHM | 2022 | TCRa and TCRb sequences of HLA-B*35:01/IPS-isolated T cells | https://www.ncbi.nlm.nih.gov/bioproject/PRJNA891934 | NCBI BioProject, PRJNA891934 |

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
