## [Editor Report]

This study examines the possibility that high incidence of SARS-CoV2 reactive T cells occur in apparently COVID 19 naive individuals. The study sheds new light on how unrelated virus-specific T cells might be involved in generating immunity to SARS-CoV-2 and is an elegantly performed study.

---

## [Decision Letter]

**Decision letter after peer review:**

Thank you for submitting your article "SARS-CoV-2-specific CD4^+^ and CD8^+^ T cell responses can originate from cross-reactive CMV-specific T cells" for consideration by *eLife*. Your article has been reviewed by 3 peer reviewers, and the evaluation has been overseen by a Reviewing Editor and Satyajit Rath as the Senior Editor. The reviewers have opted to remain anonymous.

Essential revisions:

1) Clarification of experimental explanations as guided by the reviewers (esp related to Reviewer 3).

2) Improvement of clarity of language/expression (esp related to Reviewer 2).

*Reviewer #1 (Recommendations for the authors):*

This is a very nice story.

Multiple potential epitopes were examined in SARS-COV2 while in CMV, only the most immunogenic antigen, pp65, was examined. The rationale for this selection is not entirely clear.

The presence of IPS/FVS cross-reactive T cells were examined in severe COVID patients. The patient cohort is very small and while 2/2 patients showed evidence of these cells, it is difficult to interpret given the limited dataset. Has this feature been examined in a larger cohort of patients?

Figures: The data in a number of the flow cytometry plots is almost impossible to read given the very small font size. The figures would be enhanced by increasing the font size of all percentages of the populations of interest.

*Reviewer #2 (Recommendations for the authors):*

Abstract: it should be made clear that the crossreactive T cells reduced viral spreading at lower MOIs.

Introduction, line 55: there are now some papers showing that mutations can result in T-cell escape so it would be good to reference these for completion.

Introduction, line 60: it may read better if this sentence read as follows: 'This finding indicates that T cells which were initially primed against other pathogens are able to cross-recognize SARS-CoV-2 antigen'.

Introduction, line 63: The sentence which starts 'Highly homologous…..' would benefit from editing as it is confusing to use the word 'sequences' twice without specifying whether this refers to DNA, RNA, peptides etc…

Introduction, line 78: suggest changing 'a potential trigger' to 'the initial trigger'.

Introduction, line 84: suggest using 'trigger' instead of 'potential source'.

Introduction, line 88: can you clarify the previously proposed role for these cross-reactive T cells?

Introduction, line 89: change 'indicated' into 'indicate', change 'COVID-19' into 'COVID-19 immunity'.

Introduction, line 91: change 'Taking' into 'Taken'.

Introduction, line 91-92: suggest deleting 'a memory response against CMV' and replacing with 'the CMV specific memory population'.

Introduction, line 98: suggest changing 'in' to 'presented by'.

Introduction, line 108: suggest changing the end of the sentence to 'preventing SARS-CoV-2 infection or reducing the severity of COVID-19'.

Results, line 123: the text states that no marked increase in CD4 T cell responses were observed after stimulation with spike in CMV+ donors. However, Figure 1C shows 3 circles depicting a CD4 T cell response to S in CMV+ donors. Please check if this is correct and alter either the text or figure accordingly.

Results, line 124: the text states that 7 CMV+ donors showed CD4 T cell responses to M. However, only three circles are visible in Figure 1C depicting CD4 T cell responses against M in CMV+ donors. Please check is this correct and if necessary correct figure or text accordingly.

Results, line: 174: need to provide an explanation for how the pp65 mapping described in Figure S2B was performed e.g. what do the H and V peptide pools refer to?

Results, line 180: why does the membrane protein epitope identity remain unidentified?

Results, general comment: it would be good to clarify why a different approach to CMV epitope mapping was taken for CD4 versus CD8 T cells?

Results, line 229: clarify how these two residues are similar.

Results, line 278: clarify how it is possible to tetramer sort cells if they are below the background level.

Results, line 321: need to state 'MOI of 0.05' at the end of this sentence.

Data generated using acute infection samples: more details need to be provided for these samples (e.g. some basic information such as age, gender, time post PCR+ test, symptoms etc.). The text states that IPS/FVS specific T cells showed lower expression of CD38 and HLA-DR than IPS specific T cells. This seems to be true for donor CHZ but not for donor KDH – please check and edit as necessary. As only two samples were studied then these data should not be overinterpreted e.g. unless more donors are added to this analysis then I would not suggest referring to these data in the abstract. I would also consider deleting or editing the end of the sentence at the end of the first paragraph in the Discussion section.

Methods

Line 472: include more details of the S, S1 and S+ pools used in this study.

Figures

Figure 1A: would be good to label this so that it is clear that it refers to CD4 T cells.

Figure 1B: would be good to label this so that it is clear that it refers to CD8 T cells.

Figure 1C and 1E: some stats comparing CMV+ versus CMV- donors might be useful here (either in the figure or in the legend).

Figure 2A and B: please check stats here as I would usually expect this test to be used for multiple repeats using the same clone (rather than differing numbers of repeats with different clones).

Figure 5, legend, line 344: was does 'were taken along' mean?

Figure 5, legend, line 344: insert 'at' before '24'.

Figure 5, legend, line 349: which peptide pool was used here?

Supp Figures

Supp Figure 3: change label showing the CPL data to panel 'D'.

Supp Figure 3E: why are there different affinity values shown for peptides IPSINVHHY and LEQIKTHWL?

Supp Figure 4: make it clear in the title of this figure that this is structural modelling.

---

## [Author Response]

Reviewer #1 (Recommendations for the authors):This is a very nice story.Multiple potential epitopes were examined in SARS-COV2 while in CMV, only the most immunogenic antigen, pp65, was examined. The rationale for this selection is not entirely clear.

The main aim of the experiments displayed in figure 1 was to identify SARS-CoV-2-specific T cells in CMV seropositive and seronegative donors. We therefore focused on a broader SARS-CoV-2 peptide pool repertoire to facilitate detection of cross-reactive T cells. Pp65 was added to the experiment to confirm that the CMV seropositive donors had developed CMV-specific T cells. We understand that it was unclear why pp65 only was tested, therefore we have clarified this in the manuscript (line 124-126).

The presence of IPS/FVS cross-reactive T cells were examined in severe COVID patients. The patient cohort is very small and while 2/2 patients showed evidence of these cells, it is difficult to interpret given the limited dataset. Has this feature been examined in a larger cohort of patients?

We agree that 2 patients is limiting and ideally the cohort would include more patients. Unfortunately, we had several inclusion criteria that complicated the search for suitable samples (1) Patients had to be HLA typed which is expensive and therefore not routinely performed, (2) If HLA was known, the donor had to be HLA-B*35:01, (3) the donor had to be CMV seropositive, (4) sampling should have been performed during acute disease, (5) patient should not receive immunosuppressive treatment during or prior to sampling which is a common treatment (prednisolone and dexamethasone) for severe COVID-19. As a consequence, suitable samples to study the role of the IPS/FVS-specific T cells are, unfortunately, scarce.

Figures: The data in a number of the flow cytometry plots is almost impossible to read given the very small font size. The figures would be enhanced by increasing the font size of all percentages of the populations of interest.

We apologize for the inconvenience, the percentages are now in larger font size or deleted when not relevant.

Reviewer #2 (Recommendations for the authors):Abstract: it should be made clear that the crossreactive T cells reduced viral spreading at lower MOIs.

We agree that the abstract should accurately describe the conclusions in the specific context of the study. The sentence has been completely removed following the comments by the other reviewers and our own views after re-evaluating the abstract (line 37-39).

Introduction, line 55: there are now some papers showing that mutations can result in T-cell escape so it would be good to reference these for completion.

The majority of the literature investigating this subject describes that the T cell response is minimally affected by mutations found in the current variants of concern, whilst antibodies are affected.[1-9] We therefore believe that the sentence in the original manuscript is still in line with the consensus of the current data. To update the introduction, we added “or vaccination” (line 55) and cited new literature (lines 55-58).

Introduction, line 60: it may read better if this sentence read as follows: 'This finding indicates that T cells which were initially primed against other pathogens are able to cross-recognize SARS-CoV-2 antigen'.

Thank you for this suggestion, this has been adapted in the manuscript (lines 62-63).

Introduction, line 63: The sentence which starts 'Highly homologous…..' would benefit from editing as it is confusing to use the word 'sequences' twice without specifying whether this refers to DNA, RNA, peptides etc…

Thank you for this suggestion, this has been adapted in the manuscript (lines 64-65).

Introduction, line 78: suggest changing 'a potential trigger' to 'the initial trigger'.

Thank you for this suggestion, this has been adapted in the manuscript (line 80).

Introduction, line 84: suggest using 'trigger' instead of 'potential source'.

Thank you for this suggestion, this has been adapted in the manuscript (line 86).

Introduction, line 88: can you clarify the previously proposed role for these cross-reactive T cells?

We removed part of the sentence as we see that it was a confusing sentence (lines 90-91). The role of cross-reactive T cells is not clear yet. Lines 88-90 describe an association between CMV and cross-reactive T cells with COVID-19. Lines 92-93 describe the current consensus on the role for cross-reactive T cells.

Introduction, line 89: change 'indicated' into 'indicate', change 'COVID-19' into 'COVID-19 immunity'.

Thank you for this suggestion, this has been adapted in the manuscript (line 92).

Introduction, line 91: change 'Taking' into 'Taken'.

Thank you for this suggestion, this has been adapted in the manuscript (line 94).

Introduction, line 91-92: suggest deleting 'a memory response against CMV' and replacing with 'the CMV specific memory population'.

Thank you for this suggestion, this has been adapted in the manuscript (lines 95).

Introduction, line 98: suggest changing 'in' to 'presented by'.

Thank you for this suggestion, this has been adapted in the manuscript (line 101 and 102).

Introduction, line 108: suggest changing the end of the sentence to 'preventing SARS-CoV-2 infection or reducing the severity of COVID-19'.

Thank you for this suggestion, this has been adapted in the manuscript (lines 112).

Results, line 123: the text states that no marked increase in CD4 T cell responses were observed after stimulation with spike in CMV+ donors. However, Figure 1C shows 3 circles depicting a CD4 T cell response to S in CMV+ donors. Please check if this is correct and alter either the text or figure accordingly.

There was a mistake in the figure, it has been adapted in the new manuscript version (Figure 1C).

Results, line 124: the text states that 7 CMV+ donors showed CD4 T cell responses to M. However, only three circles are visible in Figure 1C depicting CD4 T cell responses against M in CMV+ donors. Please check is this correct and if necessary correct figure or text accordingly.

There was a mistake in the figure and text which has been adapted in the new manuscript version (Figure 1C and line 131).

Results, line: 174: need to provide an explanation for how the pp65 mapping described in Figure S2B was performed e.g. what do the H and V peptide pools refer to?

The method has been elaborated in the figure legend (figure S2B, Lines 1057-1058), material and methods (lines 572-576) and Results section (line 184-185).

Results, line 180: why does the membrane protein epitope identity remain unidentified?

We aimed to identify the membrane protein epitope by separately ordering the 15-mers as peptide library, high pH Reversed Phase HPLC fragmentation of the membrane peptide pool, and *in silico* prediction based on sequence similarity to the CMV pp65 epitope AGILARNLVPM. Unfortunately, none of these methods resulted in identification of the epitope. We speculate that the epitope has hydrophobic characteristics, as usually found in membrane peptides, thereby complicating synthesis (also suggested by the manufacturer) and handling. We agree that this should be explained in the text, since it is currently unclear whether we attempted to identify the epitope (line 191-192).

Results, general comment: it would be good to clarify why a different approach to CMV epitope mapping was taken for CD4 versus CD8 T cells?

The CPL assay is commonly used to identify the epitope of CD8^+^ T cell clones. However, for CD4^+^ T cells this is not the case since the CPL assay often results in recognition patterns which are too complex to interpret. This is due to complicating factors which are typical for TCRs found in CD4^+^ T cells, such as variable epitope lengths and varying overhanging amino acids. Therefore, we decided to take an hypothesis-driven approach to identify the epitope. We clarified the use of CPL assay in lines 205-206.

Results, line 229: clarify how these two residues are similar.

This has been clarified in the manuscript (line 249-250).

Results, line 278: clarify how it is possible to tetramer sort cells if they are below the background level.

For tetramer+ T cells that are measurable, but below background level, it is unclear whether these events are background or whether it is a true but small population. This was the case for JZX in figure 1C. For flow cytometry analysis (Figure 3C) 1-2x10^6^ PBMCs were used whilst for sorting 10-50 fold more PBMCs (45x10^6^ PBMCs in case of JZX) were first enriched using magnetic activated cell sorting (MACS) followed by flow activated cell sorting (FACS). This reduced the background level and increases the purity of tetramer+ cells which can also be observed in the clonality of the TCR sequencing in figure 4a. This has been clarified in line 310, and in material and methods lines 603 and 611.

Results, line 321: need to state 'MOI of 0.05' at the end of this sentence.

This has been added to the manuscript (lines 358-359).

Data generated using acute infection samples: more details need to be provided for these samples (e.g. some basic information such as age, gender, time post PCR+ test, symptoms etc.). The text states that IPS/FVS specific T cells showed lower expression of CD38 and HLA-DR than IPS specific T cells. This seems to be true for donor CHZ but not for donor KDH – please check and edit as necessary. As only two samples were studied then these data should not be overinterpreted e.g. unless more donors are added to this analysis then I would not suggest referring to these data in the abstract. I would also consider deleting or editing the end of the sentence at the end of the first paragraph in the Discussion section.

a) Indications that these patients suffered from critical disease was indeed lacking. We added the WHO classification and additional information in table S1. We do not have time post PCR+ test, but we did include time post symptom onset and ICU admission.

b) This conclusion was indeed not in line with the figure and also not the main message from the figure. We have adapted this as it is more important to note that FVS/IPS-specific T cells are not more activated than IPS-specific T cells (line 369).

c) We agree with this comment and lines 37-39 have been removed and line 414 adapted.

MethodsLine 472: include more details of the S, S1 and S+ pools used in this study.

This has been clarified in lines 123 and 529-531.

FiguresFigure 1A: would be good to label this so that it is clear that it refers to CD4 T cells.

Has been adapted in manuscript to improve understanding of the figure at first glance.

Figure 1B: would be good to label this so that it is clear that it refers to CD8 T cells.

Has been adapted in manuscript to improve understanding of the figure at first glance.

Figure 1C and 1E: some stats comparing CMV+ versus CMV- donors might be useful here (either in the figure or in the legend).

The figure was mainly focused on descriptive statistics: do we see SARS-CoV-2 specific T cells in CMV-seropositive donors that are not present in CMV-seronegative donors. Membrane-specific CD4^+^ T cell responses were not significantly increased in CMV+ compared to CMV- (Mann-Whitney *U*-test, data did not pass normality tests). Which is also what we would expect, since these 2-6 responses are more regarded as “outliers” by the test. Therefore, we cannot claim that there is a cohort-wide correlation (we show there is a mechanistic correlation) between CMV seropositivity and membrane-specific CD4^+^ T cells, since only 6 responses were detected.

Figure 2A and B: please check stats here as I would usually expect this test to be used for multiple repeats using the same clone (rather than differing numbers of repeats with different clones).

The data points were displayed this way to be transparent regarding the number of experimental repeats. However, we understand that this is an uncommon way to show the data. The figure has been adapted by plotting the mean of the experimental replicates per clone and then a paired t-test was performed to test significance. The number of experimental repeats are depicted in the figure legend.

Figure 5, legend, line 344: was does 'were taken along' mean?

This has been clarified in the manuscript (lines 384-385).

Figure 5, legend, line 344: insert 'at' before '24'.

This has been modified in the manuscript (line 388).

Figure 5, legend, line 349: which peptide pool was used here?

This has been clarified in the manuscript (line 391).

Supp FiguresSupp Figure 3: change label showing the CPL data to panel 'D'.

This has been corrected in the manuscript.

Supp Figure 3E: why are there different affinity values shown for peptides IPSINVHHY and LEQIKTHWL?

This was a copy-paste mistake from two different tables. It has been corrected.

Supp Figure 4: make it clear in the title of this figure that this is structural modelling.

This has been adapted in the manuscript (lines 262 and 264).

References

1. Jung, M.K., et al., BNT162b2-induced memory T cells respond to the Omicron variant with preserved polyfunctionality. Nature Microbiology, 2022. 7(6): p. 909-917.

2. Tarke, A., et al., SARS-CoV-2 vaccination induces immunological T cell memory able to cross-recognize variants from Α to Omicron. Cell, 2022.

3. Keeton, R., et al., T cell responses to SARS-CoV-2 spike cross-recognize Omicron. Nature, 2022.

4. Gao, Y., et al., Ancestral SARS-CoV-2-specific T cells cross-recognize the Omicron variant. Nat Med, 2022.

5. Choi, S.J., et al., T cell epitopes in SARS-CoV-2 proteins are substantially conserved in the Omicron variant. Cell Mol Immunol, 2022. 19(3): p. 447-448.

6. Redd, A.D., et al., Minimal Crossover between Mutations Associated with Omicron Variant of SARS-CoV-2 and CD8(+) T-Cell Epitopes Identified in COVID-19 Convalescent Individuals. mBio, 2022. 13(2): p. e0361721.

7. Liu, J., et al., Vaccines Elicit Highly Conserved Cellular Immunity to SARS-CoV-2 Omicron. Nature, 2022.

8. Chiuppesi, F., et al., Vaccine-induced spike- and nucleocapsid-specific cellular responses maintain potent cross-reactivity to SARS-CoV-2 Δ and Omicron variants. iScience, 2022. 25(8): p. 104745.

9. GeurtsvanKessel, C.H., et al., Divergent SARS CoV-2 Omicron-reactive T- and B cell responses in COVID-19 vaccine recipients. Sci Immunol, 2022: p. eabo2202.